# Enhancement of Image Quality in LCD by Doping γ-Fe_2_O_3_ Nanoparticles and Reducing Friction Torque Difference

**DOI:** 10.3390/nano8110911

**Published:** 2018-11-06

**Authors:** Lin Gao, Yayu Dai, Tong Li, Zongyuan Tang, Xueqian Zhao, Zhenjie Li, Xiangshen Meng, Zhenghong He, Jian Li, Minglei Cai, Xiaoyan Wang, Jiliang Zhu, Hongyu Xing, Wenjiang Ye

**Affiliations:** 1School of Sciences, Hebei University of Technology, Tianjin 300401, China; lin_gaoedu@163.com (L.G.); Yayu_Dai@163.com (Y.D.); m18131124209@163.com (T.L.); tzy160118@163.com (Z.T.); Zxq6668786218@163.com (X.Z.); lizhenjieruning@163.com (Z.L.); zjl-656969@163.com (J.Z.); 2004041@hebut.edu.cn (H.X.); 2School of Physical Science and Technology, Southwest University, Chongqing 400715, China; mxs_mf@126.com (X.M); hezhenho@swu.edu.cn (Z.H.); aizhong@swu.edu.cn (J.L.); 3Hebei Jiya Electronics Co. Ltd., Shijiazhuang 050071, China; cml@jiyalcd.com (M.C.); wwxxy@jiyalcd.com (X.W.); 4Hebei Provincial Research Center of LCD Engineering Technology, Shijiazhuang 050071, China

**Keywords:** image sticking, residual DC voltage, liquid crystal, γ-Fe_2_O_3_ nanoparticles, friction torque

## Abstract

Improving image sticking in liquid crystal display (LCD) has attracted tremendous interest because of its potential to enhance the quality of the display image. Here, we proposed a method to evaluate the residual direct current (DC) voltage by varying liquid crystal (LC) cell capacitance under the combined action of alternating current (AC) and DC signals. This method was then used to study the improvement of image sticking by doping γ-Fe_2_O_3_ nanoparticles into LC materials and adjusting the friction torque difference of the upper and lower substrates. Detailed analysis and comparison of residual characteristics for LC materials with different doping concentrations revealed that the LC material, added with 0.02 wt% γ-Fe_2_O_3_ nanoparticles, can absorb the majority of free ions stably, thereby reducing the residual DC voltage and extending the time to reach the saturated state. The physical properties of the LC materials were enhanced by the addition of a small amount of nanoparticles and the response time of doping 0.02 wt% γ-Fe_2_O_3_ nanoparticles was about 10% faster than that of pure LC. Furthermore, the lower absolute value of the friction torque difference between the upper and lower substrates contributed to the reduction of the residual DC voltage induced by ion adsorption in the LC cell under the same conditions. To promote the image quality of different display frames in the switching process, we added small amounts of the nanoparticles to the LC materials and controlled friction technology accurately to ensure the same torque. Both approaches were proven to be highly feasible.

## 1. Introduction

With the widespread use of liquid crystal display (LCD) in mobile phones, computers, and televisions, this technology is continuously undergoing improvement to achieve low power consumption, fast response, wide viewing angle, and high contrast [1,2,3,4,5]. Consequently, high image quality in thin film transistor-liquid crystal display (TFT-LCD) has several requirements. For instance, image sticking, a phenomenon where the previous pattern remains visible when the next pattern is addressed, warrants optimization. This phenomenon is caused by direct current (DC) bias voltage and ionic impurities. Under the action of DC bias, ionic impurities accumulate on the alignment layers to form residual DC voltage, which is mainly used to evaluate the severity of image sticking [6,7,8]. 

At present, the most widely studied electrical method is to obtain residual voltage by establishing an ion kinetic model to describe ion adsorption and desorption under applied DC electric field [9,10,11]. The optical method obtains the residual voltage through the conversion of the relation between voltage and transmittance [11,12]. Electrical methods do not require a conversion process to produce more accurate results than optical methods. However, the above electrical methods also have some disadvantages. For instance, a small amount of ions escape the alignment layer during testing after the removal of the applied DC voltage, resulting in low measurement values. Moreover, some parameters in the model are difficult to be determined and are greatly influenced by ambient temperature. An appropriate method to determine the residual DC voltage is important to evaluate the image sticking of LCD accurately.

Reducing or eliminating image sticking could be realized in two aspects. On the one hand, suspensions obtained using nanoscience exhibit functions not observed in their pure state: Excellent electro-optic properties, fast response time, low threshold voltage, and spontaneous orientation [13,14,15,16,17,18,19]. Recently, some papers revealed that nanomaterials in liquid crystals (LCs) could behave as either ion-capturing agents or ion-generating objects [20,21,22,23]. Therefore, the dispersion of specific nanoparticles into LC materials for the reduction of ion-related effects such as image sticking through the ion capturing effect is an attractive alternative to the conventional manufacturing process [24,25,26]. An LC cell consisting of LC doped with nanoparticles absorb impurities and consequently decrease the number of free ions [27,28,29,30]. In addition, the number of ions that accumulate in the alignment layer changes with the doping concentration under the action of different DC bias voltages. On the other hand, the alignment layer is a key factor influencing image sticking. Previous studies reduced ion adsorption and increased the voltage holding ratio in the alignment layer by changing the structure and composition of the materials [31,32]. Lowering the tilt angle and increasing the thickness of alignment layers could also improve image sticking [33]. However, further discussion about the fabrication of alignment layers, especially the friction torque differences on the upper and lower substrates, is warranted. 

In this paper, we proposed a method for evaluating residual DC voltage in an LC cell through the slow change of capacitance. Using this method, we investigated the influence of γ-Fe_2_O_3_ nanoparticles on the residual DC voltage characteristics and explored the underlying mechanism. The residual DC voltage of friction torque differences on the upper and lower substrates was also discussed.

## 2. Experimental

### 2.1. Liquid Crystal (LC) Materials

LC materials, which are prone to image sticking in practical applications, were provided by Shijiazhuang ChengzhiYonghua Display Materials Co. Ltd. (Shijiazhuang, China) The basic physical parameters of the materials at 20 °C are as follows: Clear point of 90.3 °C; moderate dielectric anisotropy of Δ*ε* = 6.6; elastic constants *k*_11_ = 13.3 pN, *k*_22_ = 6.65 pN, and *k*_33_ =15.6 pN; and birefringence of ∆*n* = 0.11 when the wavelength is 589.3 nm. γ-Fe_2_O_3_ nanoparticles were prepared by chemically induced transition [34,35,36]. For the stable and uniform mixing of nanoparticles of different concentrations in the LC material system, the nanoparticles were first coated with oleic acid and then ultrasonicated for 30 min at room temperature. The average sizes of magnetic core and oleic acid thickness in the com-nanoparticles (γ-Fe_2_O_3_/oleic acid) used in this study were 10 nm and 2 nm, respectively. Finally, we configured four doping concentrations from 0.02 wt% to 0.11 wt% at intervals of 0.03 wt%.

### 2.2. Measurement of LC Cells

The thickness of LC cell gaps was measured with an Ultraviolet–visible spectrophotometer UV-9000S (Metash, Shanghai, China). The transmittance of light at different wavelengths was obtained through the LC cell sample. According to the principle of interference, i.e., the wavelength corresponding to the maximum value when two adjacent transmittances are almost very close or even the same, the thickness of the cell was calculated through these two wavelengths. A non-contact surface profiler Contor GK-T (Bruker, Karlsruhe, Germany), which directly presents the thickness and surface smoothness, was used to acquire 3D topographical images of the alignment layer. Both tests were carried out several times, and the final corresponding thickness was determined by the average value to minimize the error as much as possible.

### 2.3. Physics Properties 

The clear point was performed using a polarized optical microscope (POM) BX51 (Olympus, Tokyo, Japan) and the temperature of LC material was controlled by the precision hot stage LTS350 (Linkam, Surrey, UK). When the samples reached the clear point, the LC molecules would change from the disordered distribution to the arrangement parallel to the substrate, which was accompanied by the refresh of the picture. To test the clear point more accurately, we controlled the temperature of the hot stage to decline at the rate of 0.1 degree per minute. The dielectric constants *ε*_//_ and *ε*_⊥_ were measured by the dual-cell method and the LC capacitance model [24]. Parallel dielectric constant was obtained by the LC layer capacitance of the vertically aligned nematic (VAN) cell for high voltage, and the vertical dielectric constant was obtained by the LC layer capacitance of parallel-aligned nematic (PAN) cell for low voltage (below the threshold voltage). The instrument used to measure the LC cell capacitance was the precision LCR meter E4980A (Agilent, Palo Alto, CA, USA). Under the applied voltage, the LC molecules in the PAN cell would only exhibit the splay deformation; the *k*_11_ was acquired from the capacitance versus voltage measurement [37].

### 2.4. Evaluation of Residual Direct Current (DC) Voltage

The residual DC voltage was evaluated by testing the variation of the capacitance of the parallel-aligned nematic (PAN) cell when DC bias is applied to a given AC signal. The instrument used to measure the LC cell capacitance was the precision LCR meter E4980A (Agilent, Palo Alto, CA, USA). First, the LCR meter was used to measure the capacitance of the PAN cell from 0 V to 20 V only with AC signal. When the DC bias voltage exists in the driving of LCD in practical applications, the ions would drift at the substrate on both sides and form residual DC voltage. Thus, we added a DC bias to simulate a practical situation on the LC cell while applying a certain AC signal. Residual DC voltage could be acquired through the slow change of the capacitance. We evaluated the residual DC as a function of time for the application of various external DC bias voltages and set the external DC bias voltage at the range of 0.2–0.8 V with an AC voltage of 2 V.

### 2.5. Dynamic Response

Figure 1 depicts the measurement setup of the dynamic response. The absorption axes of the polarizer and the analyzer were placed orthogonally on opposite sides of the LC cell. Impulse voltage with a time of 1 s and an amplitude of 2 V was applied to the PAN cell. When applying or withdrawing voltage occurred, the changes in the intensity of a laser beam (632.8 nm) that passed through the PAN cell were monitored by a detector that is connected to the oscilloscope (Tektronix MDO3024, Johnston, OH, USA). Meanwhile, the rise time and decay time were obtained directly from the oscilloscope. The temperature was controlled at 20 °C by a hot stage LTS350 (Linkam, Surrey, UK) to ensure the accuracy of the experiment. According to Reference [37], the rotational viscosity could be calculated by the response time.

## 3. Results and Discussion

### 3.1. Physics Properties of LCs

The physical properties and related parameters of LC materials, doped with γ-Fe_2_O_3_, play a vitally important role in practical application. The clear point, threshold voltage (*U*_th_), dielectric anisotropy (Δ*ε*), elastic constant of *k*_11_, and rotational viscosity (*γ*_1_) were shown in Table 1. Almost all of the above parameters show a downward trend, except the clear point, with the increase of the doping concentration. The clear points were slightly widened and did not change significantly with the doping concentration. Low threshold voltage can reduce the power consumption of LC devices, when the doping concentration is 0.11 wt%, its threshold value was reduced by 6.7%. The response time is closely related to rotational viscosity and the elastic constant of *k*_11_ in PAN cells, both had apparent decreases, especially rotational viscosity. It is clear that doping a small number of nanoparticles did not damage the fundamental physical properties of LCs, but rather helped to improve their application prospects.

Figure 2 shows the POM images of PAN cells with and without nanoparticles. There are no obvious particles that can be seen under the POM of 50 times magnification, which indicates that the nanoparticles in the LC cell do not agglomerate and the nanoparticles are uniformly mixed without perturbing LC orientation.

### 3.2. Determination of Image Sticking

A small amount of ionic impurities could be introduced into the LC cells during synthesis and manufacture. In addition, an uncertain DC bias voltage could be formed because of the limitations in pixel structure and driving mode. When the same static image is displayed for a long time, the DC bias voltage on the display device induces the ionic impurities inside the LC cell to accumulate continuously in the two alignment layers, resulting in residual DC voltage. The basic process is shown in Figure 3. Here, we evaluated the imaging sticking as a function of the residual DC voltage at different times.

As shown in Figure 4, the LC cell could be regarded as a capacitor whose capacitance value changes with the rotation of LC molecules under an applied electric field. For positive LC materials, the voltage and capacitance in the area where the capacitance value increases substantially with voltage almost presents a linear relationship. The waveform was applied, as illustrated in Figure 5, to simulate the actual driving signal. Results show that the effective voltage on the LC layer increases and the initial capacitance changes instantaneously and reaches the maximum value when the DC voltage is applied to the LC cell under the action of a certain AC signal. Meanwhile, the impurity of ions in the LC cell would produce a directional drift. The positive and negative ions would gradually accumulate on the alignment layers on both sides of the glass substrate over time. With ion aggregation, a gradually increasing internal electric field would be formed, and the external DC electric field is offset continuously, resulting in a gradual decrease in the capacitance value of the LC cell and the corresponding voltage is the residual DC voltage. 

The dielectric constant of the LC material varies with different gray scales. The time difference between signal lines to different locations could lead to signal delay, the different feedthrough voltages caused by different gray scales, and the center of positive and negative pixel voltage deviate from Vcom at display. All of the above phenomena would result in different degrees of DC bias voltage [10,38]. Hence, we studied the variation in residual DC voltage with time under different bias voltages in the LC materials. As shown in Figure 6, the residual DC voltage increases rapidly at first and reaches a saturated state after a long period of time. The ionic migration rate increases with the increase in DC field. As the ions gradually gather on the alignment layers on both sides, the internal electric field formed by the ions gradually strengthens, and the effective DC electric field weakens, thus inhibiting the movement of ions in the LC layer, and finally reaching a dynamic equilibrium state. In addition, the magnitude of residual voltage depends on the internal DC voltage. The four other LC systems, doped with nanoparticles, show similar results, but differ in the time required to reach the saturated residual DC voltage. This difference is the main reason why image sticking occurs in the LCD during the switching process after displaying a fixed image for a long time.

### 3.3. Effect and Mechanism of Doping γ-Fe_2_O_3_ Nanoparticles

Figure 7a shows the saturated residual DC voltage when different concentrations of γ-Fe_2_O_3_ nanoparticles are doped into the LC materials at different bias voltages. The saturated residual DC voltage, formed by the impurity of ions, is lower than that of other LC materials when the doping concentration is 0.02 wt%. Under such circumstances, the decrease in saturated residual DC voltage becomes more obvious with the increase in bias voltage, which is conducive to control impure ions in the LC cell. When the DC bias voltage applied on the LC cell is 0.2 V, the saturated residual voltage could be reduced by at least 10% and the DC bias gradually increases to 0.8 V, the saturated residual voltage even decreases by more than 35%. Thus, DC bias voltage exists on the LC cell within a certain range, and the LC material doped with 0.02 wt% γ-Fe_2_O_3_ nanoparticles could absorb impure ions and realize automatic regulation of residual DC voltage to different degrees.

Image sticking is closely related to the time of showing a fixed picture, and the number of ions drifting to the two substrates would accumulate gradually over time. Figure 6 shows that the residual DC voltage would reach a saturated state. For a comprehensive assessment of the residual characteristic, not only the saturated residual voltage, but also the time required to reach this state in the LC cell, should be considered. The severity of image sticking depends on how long the process takes under the same conditions. Here, we defined the time required for residual DC voltage change from the initial to 90% of the saturated time. The time required for the LC materials with different doping concentrations to reach saturated voltage under different DC bias voltages is shown in Figure 7b. The saturated time is positively correlated with the magnitude of DC bias. Interestingly, time and residual properties are highly correlated. The LC material doped with 0.02 wt% γ-Fe_2_O_3_ nanoparticles takes a longer time to reach saturated voltage under the same conditions, and the time required for other concentrations is even faster than that for pure liquid crystal. This result indicates that trace doping is more appropriate for improving image sticking than other materials in TFT-LCD. 

γ-Fe_2_O_3_ magnetic nanoparticles doped into LC materials adsorb ionic impurities, thereby reducing the number of charged ions capable of free transport, in which the ion adsorption capacity is closely related to the doping concentration [27,28]. The non-monotonic saturated residual DC voltage may be explained via two mechanisms. The first mechanism explanation takes two types of ions and aggregation-related factors into consideration and the model proposed by Garbovskiy [29,30] described adsorption and desorption phenomena. Impurity ions in the LC cell would show three different conditions as the concentration increases. First, the free ions in the LC cell dropped rapidly, relative to the initial one, until the doping concentration was 0.02 wt%. Then, there was a reversal at the minimum and the number of free ions would increase. Finally, the impure ions were not affected by the concentration of nanoparticles, but the number of ions at this time would be greater than the initial state. This theoretical model could match the experimental results quite well, and also revealed the reason why the saturated residual DC voltage has a minimum value at a doping concentration of 0.02 wt%.

The second mechanism explaining the saturated residual DC voltage is the interface characteristics of nanomaterials. When doping at low concentrations, the interaction between the nanoparticles could be neglected. Given their large specific surface area and surface binding energy, γ-Fe_2_O_3_ nanoparticles adsorb a large proportion of the impurity ions in the LC cell and stably combine together, as illustrated in Figure 8a. With the increase in doping concentration, the interaction between nanoparticles in the LC system gradually increases, leading to the agglomeration of neighboring nanoparticles. Therefore, particle size increases with increasing doping concentration. The increase in the particle size of nanoparticles could rapidly reduce their specific surface area and surface binding, thereby weakening their surface adsorption capacity and enlarging the quantity of free ions dispersed in the LC cell (Figure 8b). Consequently, the saturated residual DC voltage would decrease to a certain value and then increase as the concentration of the nanoparticles increases. The experimental results in Figure 7 are in good agreement.

The response time should also be considered to improve the image quality by using LC doped with nanoparticles. Prolonging the response time results in tailing and even other undesirable phenomena when switching the picture being displayed. The dynamic response of pure LC and the doping concentration of 0.02 wt% are described in Figure 9. Similar to the results discussed in the article in Reference [21], the rise and decay times of the LC doped with nanoparticles are uniformly increased by 10%. Therefore, the incorporation of 0.02 wt% γ-Fe_2_O_3_ nanoparticles into the LC material could not only reduce the residual DC voltage to a certain extent, but also accelerate the response time, which could be used to improve the image quality ingeniously.

### 3.4. Residual Characteristics of Friction Torque Difference 

In the previous part, we mainly studied the residual and response characteristics of LC materials doped with γ-Fe_2_O_3_ nanoparticles. In addition, the alignment layer was considered to be a major factor causing image sticking. Hence, we continued to discuss the influence of friction torque difference between two alignment layers in the LC cell on the residual characteristics, where the friction torque represents the different rubbing strengths on the upper and lower substrates. The thickness values of the LC cells are shown in Figure 10, and the dots marked on each curve are the reference points used to calculate the thickness. As shown in Figure 11, the nearly reddish region shown in the 3D morphology is the polyimide (PI) layer coated on the substrate. Figure 11 and Figure 12 revealed that alignment layers were spin-coated uniformly and smoothly. The thickness difference is very subtle, about 60 nm, and the cause of relative altitude mutation is the residue when alignment layers were removed after high temperature heating. The same PI materials were used in the four types of LC cells. Thus, the effect of the material and thickness of the alignment layers on the result could be ignored.

Table 2 shows the four types of LC cells used in our experiments. The four PAN cells with different cell gaps and friction torques are numbered 1–4. 

To explore the effect of the manufacture process of LC cells on image sticking, we tested the residual characteristics of four kinds of LC cells with different thicknesses and friction torque values under the same conditions. The results are shown in Figure 13. The saturated residual DC voltage of the LC cell with 3.8 μm thickness is higher than that of the cell with 7.4 μm thickness under the same conditions. LC cells with different thicknesses would have different arrangement states of LC molecules driven by the same AC signal, resulting in different effective dielectric constant values. Under the action of DC bias, the LC molecules are more sensitive to the electric field inside the thinner LC cells. Thus, they are more susceptible to the residual DC voltage when displayed. 

Friction orientation plays a decisive role in the orientation and anchoring of LC cells, and it exerts an inestimable effect on display quality. In the actual operation process, the pressure of the friction cloth and the speed of the friction roller may lead to different friction torques on the upper and lower substrates. Comparison of the number 1, 3 and 2, 4 LC cells is shown in Figure 13. The saturated residual DC voltage in the LC cell increases as the friction torque between the upper and lower substrates increases, and this change is hardly limited by the thickness of the LC cells. LC cells with a small friction torque difference can improve image sticking because of their low saturated residual DC voltage and slow time to reach the saturated state. 

Reducing the residual DC voltage by increasing the thickness of the LC cell is inappropriate because this method can considerably affect the response time of the LCD. In general, the friction process must be controlled and the appropriate technological process must be selected to reduce the friction torque difference between the two substrates and thus decrease the residual DC voltage.

## 4. Conclusions

We demonstrated a simple method to evaluate residual DC voltage through capacitance change in the PAN cell and further explored methods to improve image sticking from LC materials and alignment layers. The LC material added with 0.02 wt% γ-Fe_2_O_3_ nanoparticles can stably adsorb a large number of impurity ions, thereby reducing the saturated residual DC voltage and extending the time to reach the saturated state. The residual DC voltage was also determined by the friction torque difference between the upper and lower substrates, which has a lower saturated residual DC voltage with a small friction torque difference. Thus, attention should be focused on the rubbing process to reduce or eliminate the friction torque difference. Through the implementation of these two schemes, the display of higher image quality can be realized. This study can serve as a guide for improving image quality.

## Figures and Tables

**Figure 1 nanomaterials-08-00911-f001:**
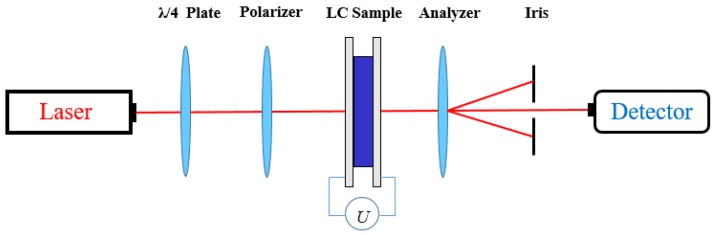
Optical path diagram of dynamic response.

**Figure 2 nanomaterials-08-00911-f002:**
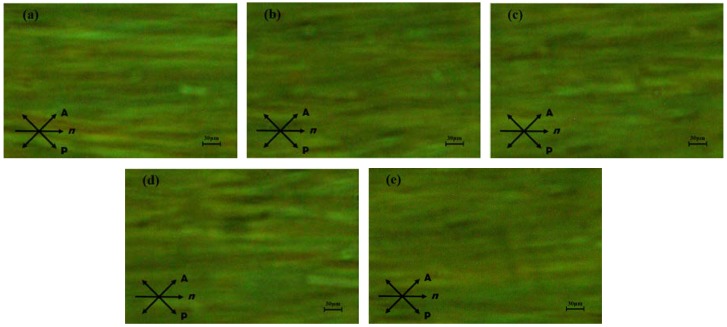
The polarized optical microscope (POM) images of parallel-aligned nematic (PAN) cell with and without γ-Fe_2_O_3_ nanoparticles. (**a**) 0.00 wt%; (**b**) 0.02 wt%; (**c**) 0.05 wt%; (**d**) 0.08 wt%; and (**e**) 0.11 wt%.

**Figure 3 nanomaterials-08-00911-f003:**
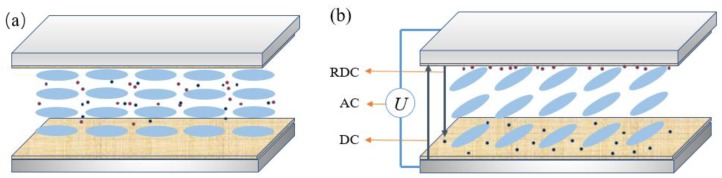
(**a**) Natural state of liquid crystal (LC) molecules and ionic impurities in the PAN cell. (**b**) Distribution of LC molecules and ionic impurities under applied voltage in the PAN cell.

**Figure 4 nanomaterials-08-00911-f004:**
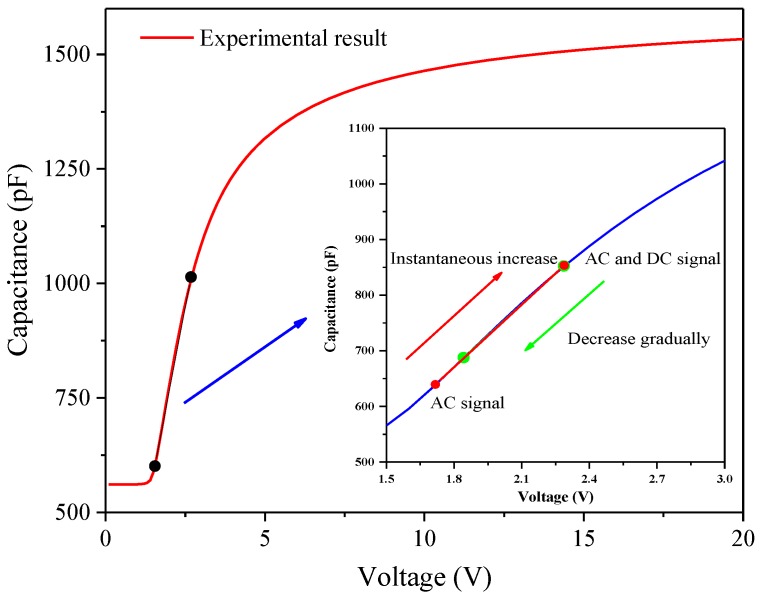
Capacitance of the LC cell varies with the voltage and the principle of evaluating residual direct current (DC) voltage.

**Figure 5 nanomaterials-08-00911-f005:**
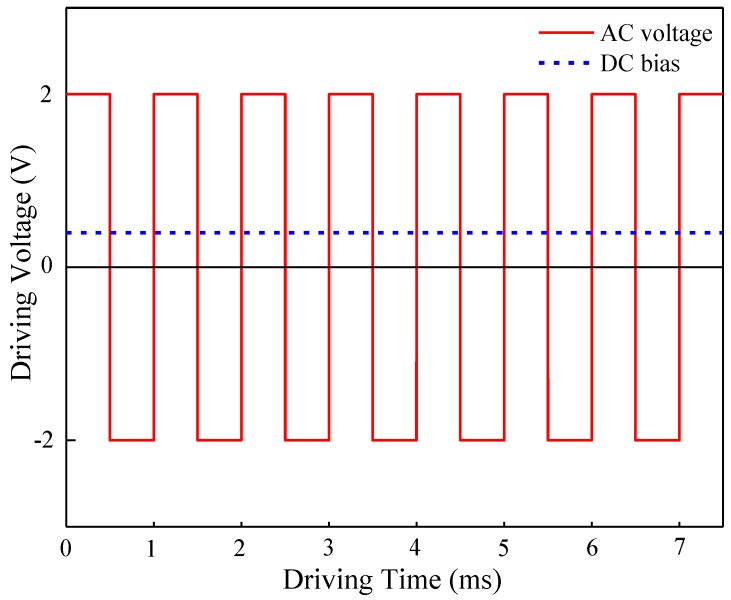
Driving voltage applied to the LC cell.

**Figure 6 nanomaterials-08-00911-f006:**
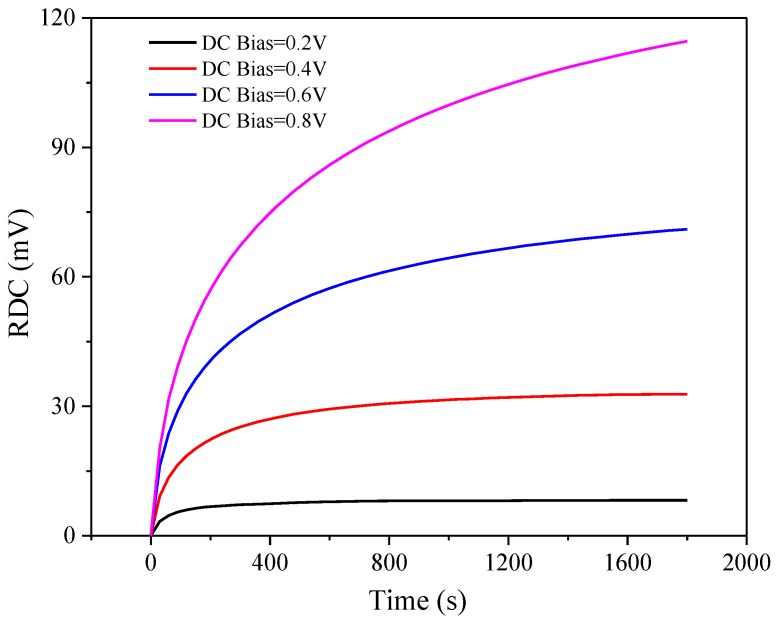
Residual DC voltage with time under different bias voltages in pure LC materials.

**Figure 7 nanomaterials-08-00911-f007:**
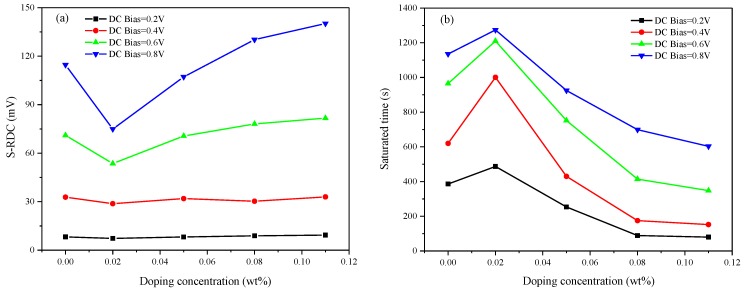
(**a**) Saturated residual DC voltage of different concentrations at different DC bias voltages. (**b**) Saturated time of different concentrations at different DC bias voltages.

**Figure 8 nanomaterials-08-00911-f008:**
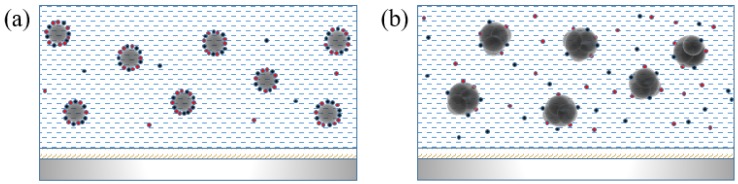
(**a**) γ-Fe_2_O_3_ nanoparticles doped at low concentrations and (**b**) γ-Fe_2_O_3_ nanoparticles doped at relatively high concentrations.

**Figure 9 nanomaterials-08-00911-f009:**
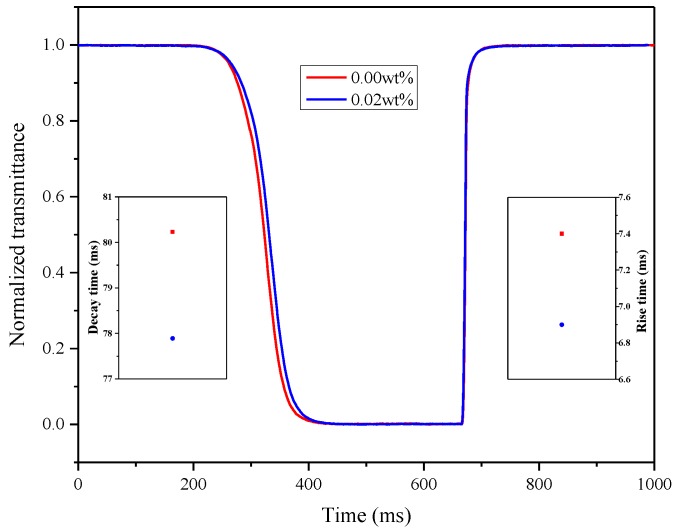
Response time of 0.00 wt% and 0.02 wt% γ-Fe_2_O_3_ nanoparticle-doped LC materials.

**Figure 10 nanomaterials-08-00911-f010:**
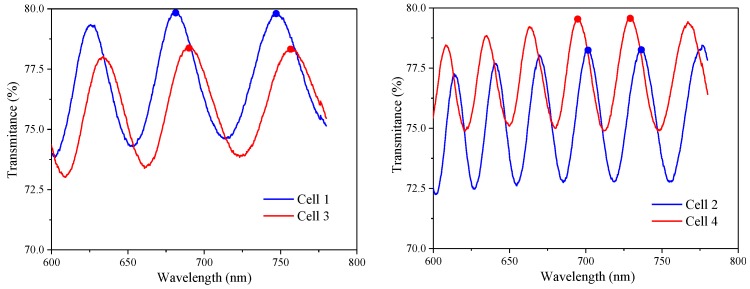
Wavelength-transmittance (λ-T) curves of four types of LC cells.

**Figure 11 nanomaterials-08-00911-f011:**
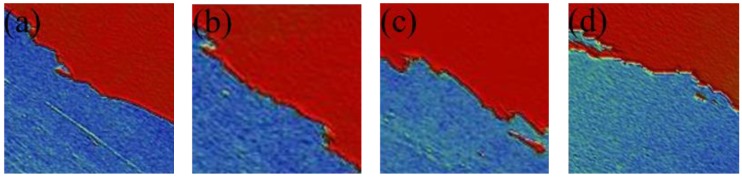
Three-dimensional topography of alignment layer of four LC cells (**a**) Cell 1; (**b**) Cell 2; (**c**) Cell 3; and (**d**) Cell 4.

**Figure 12 nanomaterials-08-00911-f012:**
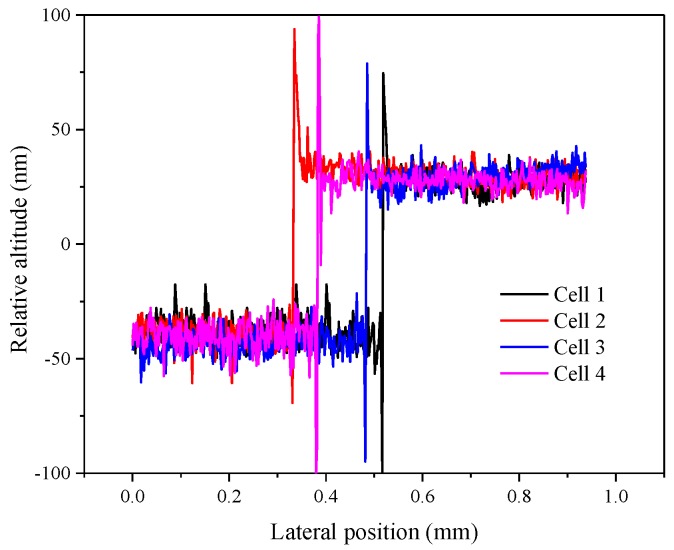
The thickness of alignment layers of four LC cells.

**Figure 13 nanomaterials-08-00911-f013:**
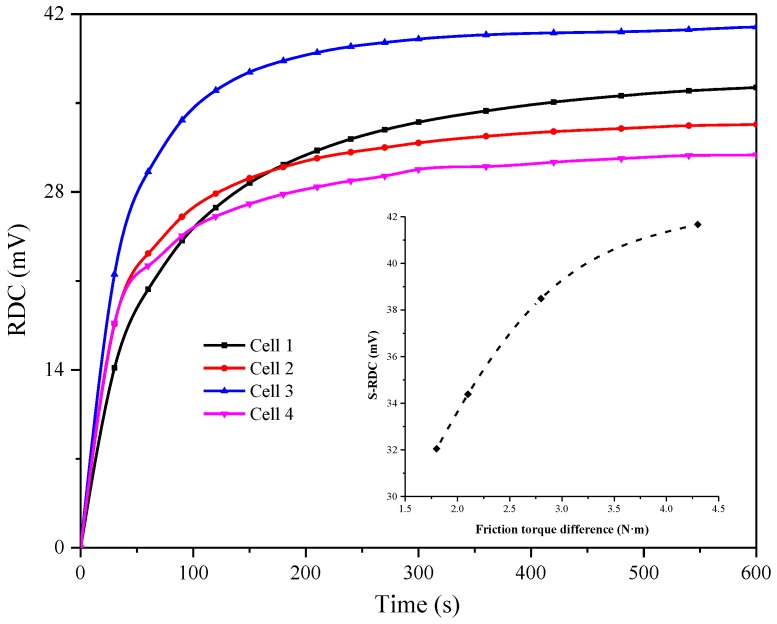
Variation in residual DC voltage with time, in which the inserted graph shows the relation between saturated residual DC voltage and friction torque difference.

**Table 1 nanomaterials-08-00911-t001:** The physics properties of liquid crystal (LC) with and without nanoparticles.

Samples	Clear Point (°C)	*U*_th_ (V)	Δ*ε*	*k*_11_ (pN)	*γ*_1_ (mpa·s)
Undoped LC	91.3	1.495	6.58	13.20	19.49
LC + 0.02 wt% γ-Fe_2_O_3_	93.6	1.463	6.67	12.81	17.97
LC + 0.05 wt% γ-Fe_2_O_3_	93.1	1.452	6.71	12.7	16.22
LC + 0.08 wt% γ-Fe_2_O_3_	92.0	1.437	6.70	12.42	15.27
LC + 0.11 wt% γ-Fe_2_O_3_	92.9	1.394	6.74	11.75	14.03

**Table 2 nanomaterials-08-00911-t002:** Parallel-aligned nematic (PAN) cells with different cell gaps and friction torques.

Number	1	2	3	4
Cell gap/μm	3.8 ± 0.1	7.4 ± 0.1	3.8 ± 0.1	7.4 ± 0.1
Thickness of PI layer/nm	60.8	61.7	61.2	60.7
Friction torque on up substrates/N·m	2.7	2.2	25.3	25.5
Friction torque on lower substrates/N·m	5.5	4.3	29.6	27.3

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
