# Peer review of "Enhancement of Image Quality in LCD by Doping γ-Fe2O3 Nanoparticles and Reducing Friction Torque Difference"

_nanomaterials, 2018, doi:10.3390/nano8110911_

Round 1

Reviewer 1 Report

The submitted paper reports experimental studies of image sticking phenomena in liquid crystal cells and the effects of γ-Fe2O3 nanoparticles on the image sticking in liquid crystals. Upon reading the paper, I have an overall positive impression of the experimental part of this manuscript. However, the discussion of the obtained experimental results is not convincing and should be improved. The following specific points should be addressed:

1) Introduction. The literature review should be improved. The authors highlight the use of nanomaterials in liquid crystals for the reduction of ion-related effects such as image sticking through the ion capturing effect. In fact, in general, nanomaterials in liquid crystals can behave as either ion-capturing objects or as ion-generating objects (for example, please see recent review and research papers such as “Nanoparticle-Enabled Ion Trapping and Ion Generation in Liquid Crystals” Advances in Condensed Matter Physics, Volume 2018, Article ID 8914891, 8 pages, https://doi.org/10.1155/2018/8914891 “ ; ““Impact of titanium dioxide nanoparticles on purification and contamination of nematic liquid crystals,” Beilstein Journal of Nanotechnology, vol. 8, pp. 2766–2770, 2017. doi:10.3762/bjnano.8.275”; Nano-Objects and Ions in Liquid Crystals: Ion Trapping Effect and Related Phenomena Crystals 2015, 5(4), 501-533; https://doi.org/10.3390/cryst5040501 ; “Nanomaterials in Liquid Crystals as Ion-Generating and Ion-Capturing Objects, Crystals 2018, 8(7), 264; https://doi.org/10.3390/cryst8070264 ” and references therein). Upon reading the introduction in its current form the reader can receive an incorrect impression that by dispersing nanomaterial sin liquid crystals we can immediately reduce the concertation of mobile ions and image sticking effect. In fact, it can be achieved only under certain conditions not discussed by the authors (and this experimental fact it was reported by many independent research groups). I suggest the authors modify their introduction to account for latest results and modern conceptions of the behaviour of nanomaterials and ions in liquid crystals. In addition, I do not think that the proposed method to evaluate residual DC voltage is totally new as claimed by the authors. In fact, this is a modification of routinely used experimental techniques.

2) Experimental methods. The authors should provide detailed description how they defined and measured the friction torque on substrates of the liquid crystal cell.

3) Experimental results and discussion.

3.1. I would not use the statement about liner dependence of the capacitance on the applied voltage. It is true only for a very narrow range of the applied voltage and is useless over a broader voltage range typically utilize din experiments.

3.2. The discussion of results shown in Figures 6, 7 is not convincing and should be improved. The observed non-monotonous dependence of both saturated residual DC voltage and effective time constant on the concentration of nanoparticles can suggest the presence of several types of ions in liquid crystals (see recent papers “ Ions and size effects in nanoparticle/liquid crystal colloids sandwiched between two substrates. The case of two types of fully ionized species, Chemical Physics Letters, Volume 679, 1 July 2017, Pages 77-85 https://doi.org/10.1016/j.cplett.2017.04.075 “ and “Impact of contaminated nanoparticles on the non-monotonous change in the concentration of mobile ions in liquid crystals”, Liquid Crystals, Volume 43, 2016 - Issue 5, Pages 664-670, https://doi.org/10.1080/02678292.2015.1133850  “). The model based on aggregation shown in Figure 7 cannot explain the observed non-monotonous behaviour (it is easy to understand because aggregation typically is characterized by longer time constant and in the discussed cases the concentration of nanodopants was relatively small and they were covered with oleic acid). I would suggest considering several types of ions and possible aggregation-related factors.

3.3 The discussion of friction torque difference should be improved. The authors should explain how they measure the aforementioned torques and provide needed information.

3.4. Figure 8. Please show error bars in insets. The difference is very negligible and error bars will be very helpful.

3.5. Figure 10. Show the size bar.

Reviewer 2 Report

The results presented by authors are quite interesting. However, some explanations such that “The residual DC voltage was also determined by the friction torque difference between the upper and lower substrates.” are not clear enough and more information is required for the paper to be published with answers on following questions.

-Unit of elastic constant is required and what wavelength was for birefringence?

-Were nanoparticles directly mixed with LC without using solvent?

-Relatively high concentration of nanoparticles was doped into LC, which may affect physical properties of LC. Authors need to confirm any change in physical properties of LC, especially in dielectric anisotropy, rotational viscosity, and Tni.

-Please provide more detail information of nanoparticles such as size and its electrical and optical properties.

-Authors need to provide POM images of PAN cell with and without nanoparticles to confirm uniform mixing of particles without perturbing LC orientation. This may explain well enough on change in electro-optics of the cell.

-According to Fig. 8, the rise time of pure LC seems to be shorter than that in the doped cell. Please check this again.

-The residual dc mainly depends on resistivity of an alignment layer not the friction talk. Please read reference papers more carefully and authors need analyze R-DC more clearly.

-In measuring R-DC, electrical method to evaluate capacitance was applied whereas the flickering removing method is well known for this. Have you compared both methods if both approaches give the same trends in data?

-Overall, if the nanoparticles can improve electro-optics of PAN cell is not convinced enough and PAN cell is not commercialized well. What if the mixture is applied to the IPS cell? It would be very nice of authors can provide the same data in the IPS cell.

Reviewer 3 Report

The paper “Enhancement of image quality in LCD by doping γ-Fe2O3 nanoparticles and reducing friction torque difference” by Lin Gao, Yayu Dai, Tong Li, Zongyuan Tang, Xueqian Zhao, Zhenjie Li1, Xiangshen Meng, Zhenghong He, Jian Li, Minglei Cai, Xiaoyan Wang, Jiliang Zhu, Hongyu Xing and Wenjiang Ye describes phenomena of high economic importance. The basis of these LC cells is the alignment of liquid crystals. Switching of the anisotropic phases is accomplished by voltage changes. All experimental procedures are thoroughly described. The influence of γ-Fe2O3 nanoparticles on the image quality was proven.  

Author Response

Dear Reviewer:

Thank you very much for your review and affirmation of our work, which will be a great inspiration to our next work.

Round 2

Reviewer 1 Report

The authors reasonably addressed the received comments by providing new important data and adding new text, figures, and references. My overall impression is very positive and I can recommend the paper for its publication assuming some minor corrections are done:

1) Add the size of nanodopants to the text of the manuscript.

2) Add the size bar to Figure 2.

3) Inset, Figure 13: change “friction torque” to “friction torque difference”.

Author Response

Dear Reviewer:

We have read carefully the decision on the manuscript (nanomaterials-379704) submitted to Nanomaterials and we are deeply appreciate to you for pointing out some important modifications needed in the paper. We have revised our manuscript, and all changes have been marked in red in the revised manuscript. The point-by-point responses are listed as the following.

We first repeat the your suggestions in black and answer them in red.

1)       Add the size of nanodopants to the text of the manuscript.

Answer: We have added nanoparticle size information to the article.

2)       Add the size bar to Figure 2.

Answer: Thanks for your advice. The size associated with POM has been added to the Figure.

3)       Inset, Figure 13: change “friction torque” to “friction torque difference”.

Answer: Thank you very much for your reminder of this detail, we have already made corrections in the manuscript.

Reviewer 2 Report

Authors addressed most of comments I concern. It is acceptable though it is hard to accept the results that the doping increases clearing temp..

Author Response

Thank you very much for reviewing and agreeing with our previous work, which will be our driving force for moving forward.